# Effect of the Acid Medium on the Synthesis of Polybenzimidazoles Using Eaton’s Reagent

**DOI:** 10.3390/polym15092130

**Published:** 2023-04-29

**Authors:** Miriam García-Vargas, Mario Rojas-Rodríguez, Joaquín Palacios-Alquisira, Lioudmila Fomina, Carla Aguilar-Lugo, Larissa Alexandrova

**Affiliations:** 1Laboratorio de Fisicoquimica Macromolecular, Posgrado Facultad de Química, Universidad Nacional Autónoma de Mexico, Circuito Exterior s/n, Ciudad Universitaria, Mexico City 04510, Mexico; miriam.garcia@iim.unam.mx (M.G.-V.); polylab1@unam.mx (J.P.-A.); 2Instituto de Investigaciones en Materiales, Universidad Nacional Autónoma de Mexico, Circuito Exterior s/n, Ciudad Universitaria, Mexico City 04510, Mexico; mariorojas.r@outlook.com (M.R.-R.); lioud@unam.mx (L.F.); 3Facultad de Química, Universidad Nacional Autónoma de Mexico, Circuito Escolar s/n, Ciudad Universitaria, Mexico City 04510, Mexico; carla.aguilar.lugo@gmail.com

**Keywords:** polybenzimidazole, Eaton reagent, superacid

## Abstract

The influence of trifluoromethanesulfonic (TFSA) superacid on conditions of the synthesis of polybenzimidazoles, such as OPBI and CF_3_PBI, was studied. It was shown that the polycondensations proceeded smoother and at lower temperatures in the presence of the TFSA in Eaton’s Reagent and that polymers of high molecular weights, and readily soluble in organic solvents, were obtained. The effect was more pronounced for CF_3_PBI, where the low reactivity monomer, 4,4′ (hexafluoroisoproylidene)bis (benzoic acid), was used. CF_3_PBI was obtained at a moderate temperature of 140 °C with no gel fraction and exhibited an inherent viscosity twice higher than the one obtained by the traditional method. In fact, the addition of TFSA allows the obtention of soluble N-phenyl substituted CF_3_PBI by direct synthesis, which had not been obtained otherwise. Thus, the use of TFSA is a good media for the synthesis of N-substituted PBIs under relatively mild conditions.

## 1. Introduction

Aromatic polybenzimidazoles (PBIs) represent a family of high-performance polymers and are characterized by its outstanding thermo-oxidative and mechanical properties, as well as its excellent chemical and moisture resistance [1,2]. These properties make them suitable for applications under harsh conditions, such as aerospace, and defense applications. However, existing difficulties in the present state of PBI manufacturing and processing technologies limit their commercialization. Until recently, they were typically used only for thermo-protective clothing and reverse osmosis membranes [1]. In the last two decades, the interest in PBIs has recommenced due to their promising properties as membrane materials for high-temperature fuel cells [3,4,5,6,7,8], gas [3,9,10,11,12,13,14], and liquid separations [15,16], which has intensified the synthesis of novel modified PBI structures. However, the severe conditions of PBI synthesis frequently restricts the possible structure variations or does not permit the formation of soluble high molecular weight polymers.

Since Vogel and Marvel reported the first PBI synthesis via high-temperature melt condensation in the early 1960s [17], other synthetic methods have been developed to obtain better processable PBIs under milder conditions [3,18,19,20,21,22,23,24,25]. Mostly, for laboratory scale production, PBIs are synthesized by direct polycondensation between an aromatic dicarboxylic acid and tetraamine in an acidic media using a polyphosphoric acid (PPA) or Eaton’s reagent (ER) at high temperatures (200 °C or more in PPA and 140 °C in ER). 

The reaction mechanism consists of the activation of the carbonyl groups by the interaction with PPA, thus, increasing the electrophilicity of the carbonyl carbon and making the reaction between aromatic acid and amine possible [26]. Additionally, the cyclization is also an acid catalyzed reaction [27], therefore, the PPA acts as a catalyst and the reaction media simultaneously. A similar mechanism was proposed for ER [28], although high viscosity and complex compositions of PPA and ER complicated the experimental mechanistic studies. Despite some advantages of ER compared to PPA, such as its lower viscosity, stronger solubilization, and cyclization capacity, its application in polycondensation is restricted by a maximum operating temperature of 140 °C, due to the decomposition of methanesulfonic acid (MSA). This may be a problem in the synthesis of PBIs when low reactivity monomers are used in the synthesis as electron deficient dicarboxylic acids, for example, 4,4′-(hexafluoroisopropylidene)bis(benzoic acid) or isophthalic acid [3,19]. For example, CF_3_PBI of a reasonable molecular weight was obtained in the PPA at 220 °C after 10 h [12]. Another important drawback of ER is its higher promotion of side acylation reactions, that frequently results in branched or crosslinked polymers [20]. 

Attempts to avoid such harsh reaction conditions have been made by a two-stage solution polycondensation via soluble poly(*o*-aminoamide) [29,30] or poly(azomethine) [31] intermediates, but the principal disadvantages of these methods were the use of precise reagents, such as acid chlorides [29,30], high purity and dry reaction conditions [29,30,31], intermedial reaction work up, and low molecular weight resulting polymers [31]. 

Therefore, developing improved methods that permit the synthesis of the aromatic processable high molecular weight PBIs still remains the actual task. Recently, we reported the synthesis of PBIs in ER but using an unusually high temperature of 180 °C [32]. It was shown that under such conditions, PBIs with high molecular weights and were soluble in organic solvents were formed in very short reaction times. The method particularly works well for monomers considered unsuitable for the synthesis of PBIs in ER under standard conditions or requiring a very long reaction time. However, the application of these conditions for the synthesis of N-substituted PBIs resulted in either low molecular weight polymers or gels [33]. Meanwhile, the introduction of bulky substitutes in the backbone chain is one of the principal strategies to lose the chain packing and, therefore, to gain some additional free volume in the polymer [34]. It should improve not only the processability of the polymer films but also their gas permeability. Both of these features are very important for the application of rigid PBIs.

Therefore, we have been continuously searching for more optimal methods which may allow for the synthesis of soluble high molecular weight PBIs of different structures. To find better conditions that would be apt for high and low reactivity monomers and not accelerate the side crosslinking reactions, we tried to modify ER using stronger trifluoromethanesulfonic acid (TFSA). The application of the superacids in other polycondensation reactions is well-known and led to the formation of linear high molecular weight polymers at lower temperatures [35,36,37]. However, until now, we found only a few articles on the use of TFSA for PBI synthesis [38,39]. Indeed, the addition of TFSA into ER resulted in PBI formation in a shorter reaction time, even using monomers of low reactivity. However, the kinetic details and formation of N-phenyl substituted PBIs have not been reported. In this work, a series of processable PBIs with high molecular weights were synthesized using ER modified with TFSA. It was shown that in the presence of a strong TFSA, the polycondensation reactions proceeded at lower temperatures, which is particularly useful for monomers of low reactivity. The evolution of the polymers’ molecular weights and the benzimidazole cycle formation during the reaction were investigated. Additionally, the formation of a processable N-phenyl substituted polybenzimidazole was possible in direct polycondensation using these reaction conditions. 

## 2. Materials and Methods

### 2.1. Materials 

The monomers, 3,3′-diaminobenzidine (DAB, 99%), 4,4′-oxybis (benzoic acid) (OBBA, 99%), and 4,4′-(hexafluoroisopropylidene) bis(benzoic acid) (HFA, 98%) were purchased from Aldrich (Toluca, Mexico) and used without any additional purification. 1,3-Dichloro-4,6-dinitrobenzene (97%), methanesulfonic acid (MSA), nitrobenzene (99%), N-methyl pyrrolidone (NMP, 98%), dimethyl sulfoxide (DMSO, 99%), N, N-dimethylacetamide (DMAc, 99%), phosphorus pentoxide (P_2_O_5_, 98%), sodium bicarbonate (NaHCO_3_), and methanol (MeOH, 98%) were supplied by Aldrich (Toluca, Mexico), while trifluoromethanesulfonic acid (TFSA) was received from Oakwood Ltd, (Los Angeles, CA, USA). MSA, TFSA and nitrobenzene were distilled under a vacuum prior to use, and the other reagents and solvents were used as received. Eaton’s reagent (ER) was prepared by mixing freshly distilled MSA with P_2_O_5_ (10:1 wt/wt) at 30 °C and kept under N_2_. Modified Eaton’s reagent was prepared by mixing MSA/TFSA in different proportions with P_2_O_5_ (total acids/P_2_O_5_ = 10/1 wt/wt).

### 2.2. Syntheses of Polymers

The PBIs were prepared by solution polycondensations using equimolar amounts of DAB and the corresponding diacid. Synthesis of OPBI was performed as follows: DAB (100 mg, 0.466 mmol) and OBBA diacid (120.5 mg, 0.466 mmol) were added into a 50 mL Schlenk flask, degassed three times using a nitrogen-vacuum cycle. Then, the flask was filled with 1.86 mL of modified-by-TFSA ER and nitrobenzene in 50/50 (*v*/*v*) proportion. The mixture was stirred at room temperature for 5–10 min until a homogeneous solution was obtained. Finally, a CaCl_2_ tramp was adapted, and the flask was placed into an oil bath preheated to a desirable temperature under a N_2_ atmosphere. The reaction was stopped when the mixture was about to stop stirring and the product was isolated immediately by pouring it into a NaHCO_3_ water solution. It was then filtered off and washed three times with deionized water and methanol and finally dried in a vacuum oven at 60 °C for approximately 6 h to a constant weight. The same procedure was performed for the synthesis of CF_3_PBI with DAB and HFA diacid, except no nitrobenzene was used in this case.

The N-phenyl substituted tetraamine, 4,6-di(N-phenylamine)-1,3-diaminebenzene (Ph-BET), was prepared as reported elsewhere [33]. N-phenyl substituted polybenzimidazole (Ph-CF_3_PBI) was obtained using this Ph-BET (100 mg, 0.344 mmol) and HFA diacid (135 mg, 0.344 mmol) in 1.86 mL of modified ER and nitrobenzene 50/50 (*v*/*v*). ER was modified with TFSA in a 50/50 (*v*/*v*) proportion of TFSA/MSA. 140 °C was maintained during the reaction, and the reaction was stopped after 100 min just before the mixture got very viscous. Figure 1 shows the synthetic methodology for PBIs studied in this work.

### 2.3. Preparation of PBI Films

The films were prepared by casting from the polymer solutions in DMSO (250 mg of polymer in 2 mL DMSO) and dried in a vacuum oven using stepwise heating, in order to avoid defects due to the fast evaporation of the solvent: 3 h at 80 °C under 30 psi pressure, then under 60 psi pressure 1 h at 100 °C followed by 5 h at 120 °C, 1 h at 150 °C, 12 h at 180 °C and 1 h at 200 °C. The thickness of the obtained films was in the range of 40–60 μm (measured by a micrometer with an accuracy of ±5 μm).

### 2.4. Characterization Methods 

The IR spectra were recorded on a Bruker Alpha ATR spectrometer (Billerica, MA, USA). ^1^H NMR spectra were performed using a Bruker Advance III HD 400 (Billerica, MA, USA) in deuterated dimethyl sulfoxide (DMSO-d_6_) solutions. Inherent viscosities (ƞ_inh_) of 0.3 g/dL polymer solutions (in DMSO) were measured at 30 °C using an Ubbelohde viscometer. The gel fraction was measured gravimetrically, the polymer solutions were filtered off, and the retained gel was left on the filter and dried until a constant weight. Here, the gel was considered as a polymer fraction which was not soluble in dipolar organic solvents such as DMSO, NMP or DMAc at 80 °C.

The thermal stability of the polymers was measured by a thermogravimetric analysis under a N_2_ atmosphere at a heating rate of 10 °C/min in a temperature range from 25 °C to 800 °C on a DuPont 2950 Thermogravimetric Analyzer, TA Instruments (New Castle, DE, USA). 

The wide-angle X-ray diffraction (WAXD) spectra of the PBI films were recorded using a Bruker D8 Advance in a continuous scan mode using CuKα1 radiation. The average d-spacing was obtained from Bragg’s equation: nλ = 2dsinθ where d is the d-spacing, θ is the scattering angle and n is an integer number related to the Bragg order. 

The molecular weights (Mw) of the polymer samples were determined using high pressure gel permeation chromatographer (GPC) Waters 717 plus autosampler, equipped with two columns: Styragel HR 4E molecular weight range from 5 × 102 to 1 × 105 and Styragel HR 5E molecular weight range from 2 × 103 to 4 × 106. DMF was used as an eluent at 50 °C with a flow rate of 1.0 mL/min; linear polymethylmethacrylate was used as a standard.

## 3. Results and Discussion

### 3.1. OPBI and CF_3_PBI Synthesis

Direct polycondensation between aromatic dicarboxylic acids and tetraamines in ER is the preferable method for the formation of PBIs in the laboratory. Compared to PPA, another commonly used solvent for PBI synthesis, ER has a higher solubilization capacity and cyclization power, therefore, the syntheses proceed faster and at lower temperatures [19,40]. Additionally, ER is less viscous which allows an easier reaction work up, but it is a better promotor of the acylation side reactions and frequently results in the formation of branched or even crosslinked macromolecules [20]. Particularly, it affects the N-substituted structures, giving an insoluble gel or incompletely-closed benzimidazole cycles [33]. Another drawback of PBI synthesis in ER is its highest usage temperature of 140 °C, which limits its application in the cases of the monomers of reduced reactivities [19,41]. In attempts to find conditions that allow for the formation of high molecular weight and processable PBIs using even N-substituted tetraamines, we have explored the use of a stronger than MSA trifluoromethanesulfonic acid (TFSA). A combination of TFSA with P_2_O_5_ was first proposed in 1972 by the inventor of ER as an efficient promotor of acylation reactions [42], but soon TFSA was substituted by MSA due to the much lower price of the latter [43]. The application of TFSA for PBI synthesis is scarcely studied; up to date we found only two publications on the subject [38,39]. Thus, it was demonstrated that the addition of TFSA resulted in a PBI, formed from DAB and isophthalic acid, with significantly higher molecular weights than the ones obtained in a standard ER [38]. Recently, the successful synthesis of a series of structurally novel PBIs using a mixture of MSA/TFSA and P_2_O_5_ was published, while similar syntheses in ER have not led to the polymers [39]. In this research we studied the influence of TFSA on the synthesis of three different PBIs such as OPBI, CF_3_PBI, and one N-phenyl-substituted PBI. 

So, we started with syntheses of OPBI and CF_3_PBI since commercially available monomers were used for their formation. TFSA was mixed in different proportions with MSA and then P_2_O_5_ was dissolved in the mixture. Details are given in Experimental. 

The influence of the acid mixture composition and temperature on the characteristics of the formed polymers are given in Table 1. As reported in the Experimental, the monomer concentration was equal in all reactions and fixed as a 4 mL of MSA/TFSA mixture per 1 mmol of DAB.

The data showed that both temperature and acid mixture compositions were essential for the syntheses. It is known that ER is an excellent reaction media for the formation of OPBI, where both tetraamine and diacid demonstrate optimal reactivity. Soluble high molecular weight OPBI of a completely cyclized structure may be obtained in less than 1 h at 140 °C in ER, practically without the formation of gel fraction [19,44]. The reaction also proceeded at 100 °C, but it needed about 10 h to form the completely cyclized OPBI with reasonable molecular weights (OPBI-9, Table 1). This regime also produced less soluble polymers and a high content of gel [44]. In the presence of TFSA, it was possible to obtain a high molecular weight OPBI at 100 °C in less than 4 h without any gel fraction (OPBI-1, Table 1). Increasing the temperature to 120 °C led to an insoluble product in 10–20 min (OPBI-4, Table 1). On the other hand, the cyclization was incomplete at temperatures lower than 100 °C even after a long reaction time: 80–90 °C and 240 min of the reaction. The optimal acid mixture composition for OBPI was MSA/TFSA = 70/30 as the additional amount of TFSA in the mixture did not improve the polymerization (compare OPBI-1 and OPBI-3, Table 1). When a smaller concentration of TFSA was used in the reaction, changes in the polymerization kinetics were not important compared to those observed in pure ER. 

Concerning the low reactive HFA diacid, in order to obtain good quality CF_3_PBI, we needed to increase the TSFA concentration and reaction temperature. High molecular weight CF_3_PBI was synthesized at 140 °C using a 50/50 MSA/TSFA mixture, where under these conditions a polymer with ƞ_inh_ = 3.2 was obtained in 3 h (CF_3_PBI-3, Table 1). Despite its high molecular weight, the polymer was readily soluble in DMSO and amidic solvents at room temperature. However, when the temperature was lowered to 120 °C, it resulted in a polymer with incompletely closed benzimidazole cycles according to ^1^H MNR analysis and a high content of insoluble gel (CF_3_PBI-2, Table 1). As was noted before, under the conditions of slow cyclization, the gel fraction increased even when the low reactive HFA monomer was involved [20,44]. The data reported here clearly confirmed such a conclusion. 

The reduction of TFSA content slowed down the polymerization rate and the resulting polymer had lower molecular weights (CF_3_PBI-1, Table 1).

Corresponding curves of the growth of the molecular weights expressed in inherent viscosity values with the reaction time for OPBI and CF_3_PBI under the reported optimal conditions are shown in Figure 1. According to Figure 1, a much slower viscosity growth was observed in the traditional ER for both polymers at the same temperature, i.e., 100 °C for OBPI and 140 °C for CF_3_PBI, than that observed for OPBI and CF_3_PBI in the modified-with-TFSA ER. The kinetic curves in the pure ER have clearly expressed an S-shape, indicating a slow growth in the beginning and a faster growth after a certain reaction time. In both cases, the polymer synthesized in ER reached significantly lower molecular weights. We also compared the kinetic curves obtained in this study with the data in ER at 140 °C and 180 °C [32,44].

Regarding OPBI, its molecular weights were less affected by the reaction conditions. The polymers reached quite similar molecular weights in pure ER at 140 and 180 °C and in the presence of TFSA at 100 °C (Figure 1A). The viscosity growth was faster in ER at 180 °C and practically similar in ER at 140 °C and in 30/70 TFSA/MSA mixture at 100 °C.

The viscosity growth was very slow for CF_3_PBI in ER at 140 °C, a much faster increase in the molecular weights was observed in ER at 180 °C. However, it was not possible to maintain such a high temperature for a longer period of time due to the notable decomposition of MSA [44]. In fact, a longer time at 180 °C resulted in an abrupt increase in the insoluble fraction. Meanwhile, CF_3_PBI polymerization was much smoother in a 50/50 TFSA/MSA media at 140 °C, where the polymer with a high molecular weight was formed fast enough without any gel. In addition, the moderate reaction temperature allowed us to discard precautions on the MSA decomposition. 

GPC traces of the polymers obtained in the presence of TFSA are given in Figure 2.

Chromatograms of OPBI samples taken at distinct reaction times demonstrate a bi-modal molecular weight distribution, as was previously reported for this polymer synthesized in ER under different temperature regimes [20,44]. Estimation of the molecular weights, according to the polymethylmethacrylate standard, of the sample taken in 60 min (a in Figure 2A) gave M_n,GPC_ = 5 × 10^4^ Da for the low molecular weight fraction and M_n,GPC_ = 200 × 10^4^ Da for the high molecular weight fraction. As polymerization proceeded, the curve shifted to the higher molecular weight values with M_n,GPC_ = 14 × 10^4^ Da and M_n,GPC_ = 420 × 10^4^ Da for the low and high molecular weight fractions correspondingly (b in Figure 2A). The bi-modal character of the chromatograms is maintained, but simultaneously a proportion of the low molecular weight fraction is also reduced. Therefore, the low molecular weight fraction was predominating in the OPBI formed at an earlier reaction time, and then the areas of the low and high molecular weight fractions became practically equal at the end of the polycondensation.

Surprisingly, the GPC curves of CF_3_PBI samples were clearly monomodal in contrast to those reported before [44] (Figure 2B). The chromatograms were shifted to a higher molecular weight area by the course of the polycondensation, from M_n,GPC_ = 80 × 10^4^ Da to M_n,GPC_ = 760 × 10^4^ Da, and got essentially narrower with a dispersity index of about 1.5. Such a monomodal distribution indicates a near absence of any undesirable lateral reactions under these conditions, meaning the optimal ratio between the growth of the polymer chains and the formation of the benzimidazole ring.

### 3.2. IR and NMR Studies

The course of the reactions was also followed by FTIR and ^1^H NMR spectroscopies. The spectra of the samples of OPBI and CF_3_PBI, taken at different reaction times, are given in Figure 3, Figure 4, Figure 5 and Figure 6. The FTIR spectra of PBIs are rather complex, as there are no clearly distinguished characteristic bands as those of the polyimides [45], for example. Generally, there are groups of bands in the region of 1400–1650 cm^−1^, due to C=C and C=N stretching together with skeletal vibrations of the ring systems, and a very broad poorly resolved band in the 2400–3500 cm^−1^ region, arisen from a complex nature of the N-H interactions [46,47]. Therefore, it was not possible to unequivocally establish a completion of the cyclization based only on the FTIR data, so a ^1^H NMR analysis was performed for this purpose.

Figure 3 shows the evolution of the ^1^H NMR spectra during the synthesis of OPBI in TFSA/MSA = 30/70 at 100 °C. As can be seen, after 120 min of the reaction (Figure 3b), the spectrum confirms the formation of a linear OPBI with completely closed imidazole cycles. Aside from the characteristic N-H signal of imidazole is observed at 13 ppm, and the signals from aromatic protons appear in the area between 7 and 8.5 ppm, in accordance with the literature [32,48,49,50], no additional signals were detected. Longer reaction time did not lead to any changes in the spectrum (Figure 3c), however, the sample taken after 60 min of the reaction demonstrated several additional signals in the spectrum (Figure 3a) meaning incomplete cyclization. 

The FTIR spectra of these samples were also different as shown in Figure 4, although one could find the characteristic bands for benzimidazole in both spectra, in the 1400–1650 cm^−1^ and 2400–3500 cm^−1^ areas as described above. The FTIR spectrum of the fully cyclized OPBI (after 240 min raction time, Figure 4b is practically identical to the OPBI spectra described before [32,49]. In comparison with the spectrum of the incompletely cyclized polymer (Figure 4a), it contains lesser amounts of bands, and the bands look better resolved. Such that, the two closely located bands may be distinguished at 1630 and 1600 cm^−1^ in the spectrum associated with C=C/C=N stretching, and the vibration of the imidazole-benzene conjugated system, respectively [22,47]. These vibrations are included within one broad peak at 1660 cm^−1^ in the spectrum (a) (Figure 4). It is important to stress that no residual carbonyl (C=O) peak between 1720 and 1650 cm^−1^, which is generally used to detect an incomplete cyclization, was observed in the spectrum of the polymer, even after a 30 min reaction time. The other characteristic benzimidazole bands, including the absorptions at 1480 and 1440 cm^−1^ that correspond to the in-plane ring vibrations of the substituted benzimidazole, at 1280 cm^−1^ attributed to the imidazole ring breathing mode, and at 800 cm^−1^ due to the heterocyclic-ring vibration and the C-H out-of-plane bending [22,47] are readily detectable in Figure 4b. These characteristic bands are also present in the spectrum 4 (a), but they are not well pronounced because of the presence of other additional vibrations. The strong bands at 1240 and 1170 cm^−1^ are assigned to the vibration of Ar–O–Ar [32,49].

In contrast to OPBI, the formation of a benzimidazole ring in CF_3_PBI occurred very fast under the polymerization conditions. As can be concluded from Figure 5, where ^1^H NMR spectra of the polymer samples taken in different reaction times are presented, CF_3_PBI had a fully cyclized structure after 40 min of the polymerization in 50/50 TFSA/MSA mixture at 140 °C. The spectra of the samples taken at 40 min (Figure 5a) and at 180 min (Figure 5b) are identical and demonstrate all the characteristic signals of CF_3_PBI: (ppm) 13.22 (1H), 8.39–8.36 (2H), 8.09 (3H), 7.83 (4H), and 7.64–7.62 (5,6H) according with the literature [12,32,51]. However, the molecular weights of the polymer increased very significantly during this period of time (See Figure 1B), therefore, it may be concluded that the growth of the polymer chain and cyclization for CF_3_PBI proceeded simultaneously under these reaction conditions. This may be a reason for the monomodal and narrow distribution of the molecular weights observed for this polymer, since the presence of the uncyclized units mostly leads to ramifications or other undesirable side reactions. 

The FTIR spectrum of CF_3_PBI (Figure 6) looks very similar to those reported earlier for this polymer with a set of the characteristic bands, including a broad one at 3500–2800 cm^−1^ due to stretching vibrations of the bounded -NH--H groups, absorptions observed in the 1650–1380 cm^−1^ region attributed to the C=C/C=N stretching and some specific vibrations of benzimidazole, and a group of bands in 1260–1150 cm^−1^ that belong to the C-F stretching [12,32,34,52]. The peaks in the spectrum are better resolved than generally reported because of the still low molecular weight of CF_3_PBI, however, they clearly confirm the benzimidazole structure.

### 3.3. Synthesis of Ph-CF_3_PBI

One of our goals was to obtain the N-substituted PBI via direct polycondensation between tetraamine and diacid. Recently, we reported the formation of the N-phenyl substituted PBI in the reaction between N-phenyl substituted tetraamine, 4,6-di(N-phenylamine)-1,3-diaminebenzene, (Ph-BET) and OBBA diacid in ER, labeled as Ph-OPBI polymer [33]. It was shown that the synthesis of the N-Ph-substituted PBIs was extremely sensitive to the reaction conditions, such as monomer concentration and temperature, because of their strong tendency for crosslinking. Although the conditions for the formation of a soluble high molecular weight Ph-OPBI were found in ER, it was impossible to obtain a similar polymer with HFA. Only an insoluble gel was produced in this case, but the cyclization was incomplete, even in the gel fraction, according to the FTIR analysis, meaning that the slow formation of a benzimidazole ring was the main reason for gelation. 

Inspired by the fast cyclization observed for CF_3_PBI in the modified-with-TFSA ER, we repeated our attempts to synthesize a N-phenyl substituted PBI using Ph-BET and HFA. The polymerization conditions were slightly modified in comparison with those applied to the synthesis of CF_3_PBI, but the soluble polymer labeled as Ph-CF3PBI with ƞ_inh_ = 0.91 was obtained. The FTIR spectrum of Ph-CF_3_PBI, depicted in Figure 7a, exhibits the main absorptions in two areas: 1660–1400 and 1250–1100 cm^−1^, that are characteristic for PBIs based on HFA. The difference observed in these regions between spectra of Ph-CF_3_PBI and CF_3_PBI should be attributed to the vibrations derived from the aryl substituent groups (spectra (a) and (b) in Figure 7). On the other hand, it is important to stress that the area of the characteristic vibrations of the bounded -NH--H groups, 3500–2800 cm^−1^, remains essentially plane in the case of Ph-CF_3_PBI, meaning the absence of -NH bonds in the polymer. Only a small absorption from the aromatic CH stretching was noted at 3100 cm^−1^. 

Apart from the description of the Ph-CF_3_PBI spectrum, it is worth mentioning that there is a distinctiveness between the FTIRs of non-substituted CF_3_PBI given in Figure 6 and Figure 7b. Although all the characteristic bands are presented in both spectra, the bands in Figure 7b are much broader, which is explained by the much higher molecular weight of this CF_3_PBI sample. Figure 6 shows the FTIR spectrum of CF_3_PBI obtained in 40 min of the polymerization, while the spectrum given in Figure 7b was taken for the sample after 180 min of the reaction and the difference in molecular weights between these samples is about two orders of magnitude. 

The ^1^H NMR study of Ph-CF_3_PBI (Figure 8) confirmed the conclusions made from the FTIR spectroscopy results. No signal from HN- in the 13–14 ppm region was found, only one broad poorly resolved peak from the aromatic protons was observed in the area between 7 and 8 ppm. Similar signals were observed for another N-phenyl substituted PBI, N-Ph-OPBI, which synthesis was recently reported in ER by our group [33]. However, it was not possible to obtain a completely closed benzimidazole structure and some other additional signals were seen in the ^1^H NMR spectrum. 

Thus, it may be concluded that the soluble N-Ph-substituted PBI based on a low reactive HFA monomer was obtained in a fast and direct one-step synthesis. As far as we are concerned, this is a first report on the synthesis of this polymer.

Since the Ph-CF_3_PBI is a new polymer structure, its basic physical properties were studied and discussed in comparison with those of already reported PBIs.

The TGA tests were performed in order to evaluate thermal stability of the newly obtained Ph-CF_3_PBI. Its TGA thermogram is shown in Figure 9 together with the CF_3_PBI thermogram. Although the Ph-CF_3_PBI demonstrated a high thermostability where the decomposition started above 400 °C, it was less thermostable than the CF_3_PBI, where the main weight loss began above 500 °C. At temperature of 500 °C Ph-CF_3_PBI already lost about 12% of the weight. 

The lower thermostability was also found for other N-phenyl substituted PBIs in contrast to the unsubstituted ones [33,53,54]. Mechanistic studies of the thermal degradation of the N-phenyl substituted PBIs and the model compounds showed that the nitrogen -phenyl bond was the weakest link in these compounds, since an appreciable amount of benzene was detected in the early stage of the degradation [54]. Therefore, we believe that from 410 to 500 °C, the Ph-CF_3_PBI lost about a half of its phenyl substituents.

Figure 10 shows the X-ray diffraction patterns for the CF_3_-containing PBIs. Both polymers exhibit a broad halo, meaning an amorphous structure. Calculating the d-spacing from the Bragg equation (nλ = 2dsinϴ), we can observe two distinctive peaks for both polymers around 5.3 Å and 3.7 Å that indicates a co-existence of the different types of chain packing arrangements within the same polymer matrix. There is no significant difference between the d-spacing of the polymers derived from the HFA diacid independently of the tetraamine used. This is rather explained by the high hindrance capacity generated by the C(CF_3_)_2_ group that drastically restricts packing, thus preventing any contribution from pendant phenyl substituents.

## 4. Conclusions

In this work, the effect of the TFSA superacid on the synthesis of PBIs was studied. It was shown that the presence of a small amount of this superacid in ER allowed to reduce polymerization temperature and the polymerizations proceeded smoother. The effect was particularly important for the low reactivity monomers, such as the HFA diacid. The study of the formation of the CF_3_PBI over time demonstrated that the benzimidazole cycle closer occurred faster than the polymer chain growth that resulted in the monomodal low dispersity molecular weight distribution. 

Finally, the TFSA N-phenyl substituted PBI based on a HFA monomer (Ph-CF_3_PBI) was obtained in the presence of the TFSA in a one-step synthesis using N-phenyl substituted tetraamine. This is the first example of a direct synthesis of such a polymer. The attempts to obtain this PBI in the polyphosphoric acid or the standard ER were not successful due to the strong tendency for acylation and the slower formation of the benzimidazole ring.

## Data Availability

Not applicable.

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
