# Peer review of "Effect of the Acid Medium on the Synthesis of Polybenzimidazoles Using Eaton’s Reagent"

_polymers, 2023, doi:10.3390/polym15092130_

Round 1
Reviewer 1 Report
This paper provides some insights on the influence of TFSA superacid on the synthesis of various types of PBI, including OPBI, CF3PBI, and N-phenyl substituted CF3PBI with a combination of Eaton reagent. The finding can attract the readers to this journal. Some comments for further revision are recommended.
1. Abstract: The abstract is too simple. It should be revised with the key findings obtained from the studies including the selected synthesis conditions for the successful synthesis of OPBI and CF3PBI. Also, the finding associated with the synthesis of N-phenyl substituted CH3PBI should be further elaborated with more technical soundness.
2. Introduction: The N-phenyl substituted CF3PBI has not been discussed on its importance such as applications and synthesis difficulties. The reactivity of the polymerization of PBI monomer with different agents can be better described in specific values for a more sound comparison.
3. Results and discussion: there are several explanations which are not tally with the tabulated data in Table 1. E.g. line 206, the author mentioned the high molecular weight OPBI was obtained at 100 C in 4 h without any gel fraction. The table showed 210 min, which is not equal to 4h. Also on line 203, it says at 100C and needed 10 h for complete cyclized OPBI, no coherent data was found in Table 1. In addition, the intention of using <5 % gel fraction should be given a better explanation as it contradicts with the authors’ explanation on no formation of gel fraction in line 202 and 206 (e.g.). This also contradicts with the statement on line 204 mentioning that the condition produced less soluble polymer and high content of gel. There is no clue on how the gel fraction was calculated.
4. The NMR spectrum should be properly labeled on the peak position with the proposed modified PBI structure placed in the spectrum for ease of reading and comparison. This is the same for the FTIR spectrum.
5. Figure 1, typo OBPI
6. Proposed mechanism of polymerization with the use of TFSA and eaton reagent should be illustrated.
7. Line 294-296, further elaboration on the chemical bond with the extra signal on NMR should be explained.
8. Figure 7, the peak difference between Ph-CF3PBI and CF3PBI should be indicated in the spectrum. For NMR, better comparison between Ph-CF3PBI with similar structure to confirm the successful synthesis of this new compound.
9. Conclusion: The author mentioned kinetic study was demonstrated, yet there are no kinetic data presented in the current manuscript.
No comment
Author Response
The authors are very grateful to the reviewer for the useful comments. You will find point to point the answer to your comments in the attached document.

Reviewer 2 Report
This article reports the synthesis of polybenzimidazoles using Eaton's reagent with different superacid medium, and studied the influence of MSA/TFSA on conditions of the synthesis of polybenzimidazoles (OPBI and CF3PBI). The manuscript presents a certain degree of creative work. Therefore, I recommend the publication of this manuscript in Polymers after revisions. The detailed comments for this manuscript are as follows.
1. Why use a gradient vacuum oven for drying? Will it cause pores to form inside the membrane? What is the purpose of the author making the film?
2. Why did the author only consider MSA/TFSA ratios when preparing OPBI? Is it okay to only add TFSA? What impact will an increase or decrease TFSA in its quantity have on the results?
3. The author should calculate the approximate range of polymer molecular weight based on polymer viscosity or GPC test results?
4. The molecular structure of the polymer should be listed in Figure 3 and Figure 5, and the position of H should correspond to the position of the NMR peak.
5. The clarity of the images in the text is insufficient, please redraw them.
The clarity of the images in the text is insufficient, please redraw them.
Author Response
The authors are very grateful to the reviewer for the useful comments. You will find point to point the answers to the comments in the attached document.
